# The cAMP Receptor Protein (CRP) of *Vibrio mimicus* Regulates Its Bacterial Growth, Type II Secretion System, Flagellum Formation, Adhesion Genes, and Virulence

**DOI:** 10.3390/ani14030437

**Published:** 2024-01-29

**Authors:** Ziqi Tian, Fei Xiang, Kun Peng, Zhenyang Qin, Yang Feng, Bowen Huang, Ping Ouyang, Xiaoli Huang, Defang Chen, Weimin Lai, Yi Geng

**Affiliations:** 1College of Veterinary Medicine, Sichuan Agricultural University, Chengdu 611130, China; tzqlkj1998@163.com (Z.T.); xfsxxn@126.com (F.X.); pengkunzxs@163.com (K.P.); qzysicau@126.com (Z.Q.); fengyang_sicau@163.com (Y.F.); borg358680477@gmail.com (B.H.); ouyang.ping@live.cn (P.O.); nwm_mm2004@163.com (W.L.); 2Agricultural and Rural Bureau of Zhongjiang County, Deyang 618100, China; 3Department of Aquaculture, Sichuan Agricultural University, Chengdu 611130, China; hxlscau@126.com (X.H.); chendf_sicau@126.com (D.C.)

**Keywords:** *Vibrio mimicus*, cAMP receptor protein, phenotype, pathogenicity, transcriptional regulation

## Abstract

**Simple Summary:**

It is known that in highly pathogenic infections, the cAMP receptor protein (CRP) frequently plays an essential regulatory role. A highly pathogenic strain of *Vibrio mimicus* SCCF01 has been isolated from yellow catfish. To investigate the role of the cAMP receptor protein in regulating SCCF01, we created a strain with a deleted *crp* gene (Δ*crp*). The results demonstrated that the expression of genes related to the bacterial type II secretion system, flagellin, adhesion, and metalloproteinase was decreased by the deletion *crp* gene. The above resulted in modifications to the morphology of the bacteria and colonies, as well as a decrease in the motility, hemolytic activity, biofilm formation, bacterial growth, and enzyme activity. Animal experiments and cytotoxicity analyses verified that *crp* played a role in *V. mimicus* pathogenicity. In conclusion, these findings clarified the biological role of the *crp* gene in *V. mimicus*, revealed the pathogenic mechanism of the microorganism, and provided a basis for effective control and prevention of *V. mimicus* infection.

**Abstract:**

*Vibrio mimicus* is a serious pathogen in aquatic animals, resulting in significant economic losses. The cAMP receptor protein (CRP) often acts as a central regulator in highly pathogenic pathogens. *V. mimicus* SCCF01 is a highly pathogenic strain isolated from yellow catfish; the *crp* gene deletion strain (Δ*crp*) was constructed by natural transformation to determine whether this deletion affects the virulence phenotypes. Their potential molecular connections were revealed by qRT-PCR analysis. Our results showed that the absence of the *crp* gene resulted in bacterial and colony morphological changes alongside decreases in bacterial growth, hemolytic activity, biofilm formation, enzymatic activity, motility, and cell adhesion. A cell cytotoxicity assay and animal experiments confirmed that *crp* contributes to *V. mimicus* pathogenicity, as the LD_50_ of the Δ*crp* strain was 73.1-fold lower compared to the WT strain. Moreover, qRT-PCR analysis revealed the inhibition of type II secretion system genes, flagellum genes, adhesion genes, and metalloproteinase genes in the deletion strain. This resulted in the virulence phenotype differences described above. Together, these data demonstrate that the *crp* gene plays a core regulatory role in *V. mimicus* virulence and pathogenicity.

## 1. Introduction

*Vibrio mimicus* is a bacterium with a rod-shaped morphology. It was originally discovered in human diarrheal stools and ear infections and was previously classified as a non-pathogenic variant of *Vibrio cholerae* [1]. *V. mimicus* can infect various aquatic animals, including Koi carp (*Cyprinus carpio*) [2], prawns (*Penaeus vannamei*) [3], and oysters [4]. It poses a significant threat to the safety of aquatic food products. Consumption of infected aquatic products has been found to result in gastroenteritis in humans, which is characterized by symptoms such as diarrhea, nausea, vomiting, and fever [5]. In the past few years, the aquaculture industry has experienced significant economic losses due to the impact of *V. mimicus*, particularly in Siluriformes farmhouses [6,7,8]. Moreover, a highly pathogenic strain SCCF01 was identified in yellow catfish (*Pelteobagrus fulvidraco*) in China. This strain exhibits distinct epidemiological characteristics, including a short disease duration, and has been found to cause nearly 100% mortality in yellow catfish [7,9].

The cAMP receptor protein (Crp) is an allosteric protein that can bind with the second messenger cyclic 3′, 5′-AMP (cAMP). This interaction forms an archetypal bacterial global transcriptional regulator [10]. Crp could enhance the binding and transcription-initiation ability of the RNA polymerase holoenzyme for specific gene sets. An extensively researched strain of *Escherichia coli* is known to possess over 100 operons and around 500 genes that are regulated by Crp-cAMP [11]. Crp is also called catabolite gene activator protein (CAP) [12]; in cases where a preferred carbon source, such as glucose, is absent, the Crp-cAMP complex has the ability to activate certain catabolic pathways at the transcriptional level, facilitating growth on alternate substrates [10]. Furthermore, since the Crp-cAMP complex plays a significant role in the control of virulence gene expression [13], strains carrying mutations in the *crp* gene have shown reduced virulence and have been considered as promising candidates for vaccine development. This has been demonstrated in various pathogens, including *Yersinia pestis* [14], *Pasteurella multocida* [15], and *Edwardsiella ictalurid* [16]. In addition to its well-established role in regulating virulence factors, the Crp-cAMP complex has been implicated in regulating various activities, including motility ability [17], biofilm formation [18], host colonization [19], pheromone signaling [20], natural competence [21], osmotolerance [22], lipopolysaccharide modifications [23], resistance to bacteriophages [24], and integrase activity [25]. The *crp* gene has been identified in whole genome sequencing of SCCF01 by our laboratory [26]. Nevertheless, the current knowledge of *V. mimicus* is still limited. Therefore, this work aimed to examine the impact of *crp* gene deletion on the physiology and pathogenicity of *V. mimicus*. Additionally, we sought to reveal potential molecular associations between *crp* and other relevant genes.

## 2. Material and Methods

### 2.1. Bacterial Strains, Plasmids, and Growth Conditions

*V. mimicus* strain SCCF01 (WT), the highly virulent strain mentioned above, was focused on in this study. Both the plasmid pKD4 (Miaolingbio Co., Ltd., Wuhan, China) with a kanamycin (Kan) cassette flanked by FRT sites and the plasmid pCP20 (Miaolingbio Co., Ltd., Wuhan, China) with FLP recombinase were used to construct the deletion strain [27,28], and plasmid pKD46 (Miaolingbio Co., Ltd., Wuhan, China) was used to provide λ-RED recombinase [27]. Plasmid pBAD24 (Biofeng Biotech Co., Ltd., Shanghai, China) was used as an arabinose-inducible expression vector for the *crp* gene for a complementation experiment [29]. *V. mimicus* was routinely cultured in Luria–Bertani (LB) medium (Beijing Solarbio Science & Technology Co., Ltd., Beijing, China) at 28 °C. *E. coli* DH5α was used for harvesting mutant plasmid and was cultured in LB broth at 37 °C. Dulbecco’s Modified Eagle Medium (DMEM) was used for the growth of channel catfish kidney cells (CCK; Courtesy of Yangtze River Fisheries Research Institute, Chinese Academy of Fishery Sciences). Antibiotics were used at the following concentration: 100 μg/mL of ampicillin and kanamycin sulfate (Sangon Biotech (Shanghai) Co., Ltd., Shanghai, China). In order to activate the pBAD24, a supplementation of 0.2% arabinose was provided when required.

### 2.2. Construction of Deletion and Complementation Strains

*Crp* knockout was performed by natural transformation using a λ-RED recombination system based on Yu’s method with slight modifications [30]. Initially, PCR fragments were produced by combining three distinct PCR reactions corresponding to the regions, including the sequence intended for deletion and the FRT-flanked antibiotic resistance cassette. The primer information is listed in Appendix A. The pKD46 plasmid was electro-transformed into the wild-type strain. Subsequently, the produced PCR fragment was added to the competent cells of the wild-type strain with pKD46. The cell suspension was plated onto LB-agar plates containing antibiotics in order to select transformants. Furthermore, the pCP20 plasmid was electro-transformed into transformants in order to eliminate the antibiotic cassette; the flp recombinase on the plasmid was used to induce recombination at the FRT locations.

As previously stated, the *crp* gene fragment (approximately 651 bp) was amplified by using primers C-*crp*-F (*EcoRI*)/C-*crp*-R (*HindIII*), shown in Appendix A. Subsequently, the amplified fragment was inserted into the plasmid pBAD24 to create the expression plasmid pBAD-*crp*. pBAD-*crp* was first transformed into the *E. coli* DH5α for propagation, then electro-transformed into the deletion strain Δ*crp* to construct the complementary strain C-*crp*. The PCR and RT-PCR primers were designed to detect the *crp* gene external region and the normal expression of the *crp* gene, respectively (Appendix A).

### 2.3. Growth Assay

All strains (WT, Δ*crp*, and C-*crp*) were cultured at 28 °C overnight. The culture of each strain was diluted or concentrated with fresh LB medium to OD_600_ = 1. Following the Gram staining procedure, the bacteria was added to 100 mL of prepared LB medium at a concentration of 1% (*v*/*v*). The culture was then incubated in a shaker at 28 °C and 180 rpm. Absorbance measurements, with the optical density set at 600 nm, were conducted hourly over a period of 24 h utilizing a spectrophotometer. All experiments were repeated three times.

### 2.4. Hemolytic Activity Assay

Strains of SCCF01, Δ*crp*, and C-*crp* were cultivated on LB at 28 °C for 16 h. A spectrophotometer was used to adjust the bacterial count to OD_600_ = 1. The cultures were incubated on 5% sheep blood agar (SBA) plates at a temperature of 28 °C. After 96 h of incubation, the hemolytic activity was determined by measuring the diameter of the clear colorless region surrounding the colonies. Then, the protein of the three strains were extracted using Bacterial Activity-Keeping Lysis Buffer (Sangon Biotech (Shanghai) Co., Ltd.), and 100 μL of protein were added to each Oxford cup on 5% SBA for 24 h culture at 28 °C to test the hemolytic activity. The test was conducted three times independently, with each trial being completed in triplicate.

### 2.5. Congo Red Binding Assays

Congo red (CR) staining was used to determine the amount of extracellular polysaccharide produced by the different strains. A total of 5 μL of bacterial solution was added to a LB solid plate (the final concentration was 40 μg/mL CR), dried at room temperature, and cultured at 28 °C. The colony morphology was observed and photographed 4 days later. The test was conducted three times independently, with each trial being completed in triplicate.

### 2.6. Microtiter Dish Biofilm Formation Assay

The microplate detection technique was then carried out in accordance with the previously stated approach, which had been modified and coupled with the preset biofilm production conditions of *V. mimicus* in the laboratory [31]. The bacterial solution for the deletion strain was seeded into 96-well plates 8 h ahead of the wild strain and allowed to incubate at 28 °C. Following incubation, the three strains’ OD_600_ values were identical, and the negative control wells contained only medium. To get rid of the planktonic bacteria, the supernatant was taken out and cleaned with phosphate buffer saline (PBS). After fixing the attached bacteria for 15 min with methanol, the bacteria were allowed to dry for 10 min. The bacteria were dyed for 10 min using a 2% crystal violet (Sigma-Aldrich, St. Louis, MO, USA) dye solution, followed by the previously described washing and drying steps. The test was conducted three times independently, with each trial being completed in triplicate.

### 2.7. Swimming Motility Assay and Transmission Electron Microscopy (TEM) Observation

As previously reported, three strains (wild-strain SCCF01, Δ*crp*, and C-*crp*) were identified in the swimming motility experiment. A sterile toothpick was used to inoculate three equal-density bacteria into the LB medium with 0.3% ager. The bacteria were then grown for 24 h at 28 °C. The diameter of the motility halo was used to measure swimming motility. Each test was run in three separate assays, each in triplicate.

The bacterial samples were taken from the motility agar plates in order to promote flagella production. Transmission electron microscopy (TEM; JEOL JEM-1400FLASH, Tokyo, Japan) was used to examine the morphology of the bacteria using the negative staining approach. Formvar-coated copper grids were treated with 10 μL of each strain’s resuspension, and they were then negatively stained with 1% phosphotungstic acid for 1 min.

### 2.8. Cell Adhesion Assay

The CCK cells were subcultured and enumerated at 25 °C, and their cell density was adjusted to 1 × 10^6^ cells/well before being inoculated onto a 12-well cell culture plate. Then, bacteria were washed three times with PBS and then diluted with DMEM; 1 × 10^8^ CFU bacteria were added to each well at a multiplicity of infection (MOI) of 100:1, and incubated for 30 min at 28 °C. After washing the nonadherent bacteria with PBS, the bacteria and monolayers adhered to the cells were removed with the addition of 1% Triton X-100 (Beyotime Biotechnology, Shanghai, China). Following dilution, the recovered bacteria were cultivated on LB plates. Bacterial adhesion rate = (number of colonies × dilution ratio)/1 × 10^8^. The test was conducted three times independently, with each trial being completed in triplicate.

### 2.9. Enzymatic Activity Assay

The extracellular products (ECPs) of wild-type SCCF01, Δ*crp*, and complementary strain C-*crp* were prepared using the previous method with appropriate modification [32]. The total protein concentration of the ECPs was measured using the BCA Protein Assay Kit (Beijing Solarbio Science & Technology Co., Ltd.). Lecithinase, protease, urease, and gelatinase activity in the ECPs was assayed against agar plates containing 2% lecithin, 2% protein, 2% urea, and 1% gelatin, respectively. The test was conducted three times independently, with each trial being completed in triplicate.

### 2.10. Cell Cytotoxicity Assay

A Cell Counting Kit-8 (CCK-8) (Sangon Biotech (Shanghai) Co., Ltd.) was used to determine cell cytotoxicity. 1.0 × 10^5^ cultured CCK cells were seeded into each well of 96-well plates, and the cells were incubated for 24 h at 25 °C with 0.5 mg/mL ECPs of *V. mimicus* SCCF01, Δ*crp*, and C-*crp*. After incubation, 10μL CCK-8 solution containing WST-8 was added to each well and left at 37 °C for 30 min; the OD_450_ value was measured in each well by a microplate reader and the survival rate of CCK cells was calculated according to the formula. The test was conducted three times independently, with each trial being completed in triplicate. The formula used to compute the relative cell cytotoxicity is as follows:*Cell viability* = (*As* − *Ab*)/(*Ac* − *Ab*) × 100%(1)
where *As* = the absorbance of the experimental wells, *Ac* = the absorbance of the control wells, and *Ab* = the absorbance of the blank wells.

### 2.11. Pathogenicity Assay in the Catfish Model

Both the original (*V. mimicus* SCCF01) and the deletion (Δ*crp*) strain were grown under the conditions outlined above. The bacteria were washed and then diluted before the colony count and concentration were determined. Each tank contained twenty hybrid catfish (*Silurus soldatovi meridionalis*, Chen ♂ × *Silurus asotus*, Linnaeus ♀), which were divided into eleven equal groups. Before injecting 200 μL of bacterial solution into each catfish, each strain was diluted ten times with PBS and five gradients were established. An equivalent dosage of PBS was administered to the negative control. They were examined every day for two weeks, during which time any dead or moribund fish were removed, and their condition was visually assessed. Finally, Kou’s law was used to determine the median lethal dosage (LD_50_) for catfish.

### 2.12. Quantitative Reverse Transcription Polymerase Chain Reaction (qRT-PCR) Analysis

To extract the total RNA of the three different strains of bacteria (wild-type SCCF01, Δ*crp*, and C-*crp*), a Total RNA Extraction Kit (FOREGENE Biotech Chengdu Co., Ltd., Chengdu, China) was used. Subsequently, to determine the total RNA integrity and density, an RT reagent Kit (Vazyme Biotech Nanjing Co., Ltd.) was used for reverse transcription.

In a fluorescent quantitative PCR (q-PCR) system, 10 μL of ChamQ Universal SYBR qPCR Master Mix (Vazyme Biotech Nanjing Co., Ltd., Nanjing, China), 1.6 μL of diluted cDNA, 0.4 μL of each primer, and 7.6 μL of nuclease-free water were used in a 20 μL reaction. The reaction protocol consisted of 2 min at 95 °C, 40 cycles of 10 s at 95 °C, and 30 s at 60 °C. Every sample was examined three times. The 2^−ΔΔCT^ technique was used to determine the relative expression of genes; the *16S* gene was chosen as a standardized internal reference. The virulence factor primers related to the phenotype experiments were designed according to the complete genome data of *V. mimicus* SCCF01 available in our laboratory. Primers are listed in Appendix A.

### 2.13. Statistical Analysis

The SPSS (IBM SPSS Inc., Chicago, IL, USA) v.16.0 (calculation of LD_50_) and GraphPad Prism (San Diego, CA, USA) version 7.0 (basic *t*-test analysis) software programs were used for the statistical study. The results of at least three independent investigations are expressed as mean values ± standard deviations (SD). Significant differences are defined as *p* < 0.05 (referred to by *), *p* < 0.01 (referred to by **), *p* < 0.001 (referred to by ***), and *p* < 0.0001 (referred to by ****).

## 3. Results

### 3.1. Construction and Detection of the V. mimicus Deletion Strain Δcrp and Complementary Strain C-crp

To determine successful Δ*crp* and C-*crp* construction, gene expression was confirmed by RT-PCR and PCR detection, respectively. In order to examine the function of the *crp* gene in *V. mimicus*, Δ*crp* was created by removing a 633 bp segment. The confirmation of the deletion strain was achieved by PCR, which resulted in the generation of amplicons measuring 381 and 995 bp for the Δ*crp* and WT strains, respectively (Figure 1A). In addition, the complete deletion strain construction process is shown in Appendix A. The occurrence of the desired *crp* gene deletion through homologous recombination was confirmed by DNA sequencing. The analysis of the steady state mRNA levels of the *crp* gene revealed that the region was effectively eliminated in Δ*crp* and restored in C-*crp*, as demonstrated in Figure 1B.

### 3.2. Effect of crp Deletion on Growth and Morphology

A comparison of the growth-curves of the Δ*crp*, C-*crp*, and WT strains cultured in LB medium at 28 °C revealed a significant difference after 24 h (Figure 2A). At OD_600_ = 1, the CFU of the parent strain was 2.33 × 10^10^ CFU/mL, while Δ*crp* was measured at 4.15 × 10^9^ CFU/mL. This suggests that the deletion of *crp* affected the counts of bacteria. Moreover, Gram staining showed that the size of Δ*crp* was 2.15-fold larger than the wild strain (Figure 2B). This indicates that deletion of *crp* affects bacterial size.

### 3.3. Effect of crp Deletion on Type II Secretion System

The Type II secretion system (T2SS) is responsible for the secretion of a variety of extracellular proteases and toxins. We first detected hemolysin, one of the toxins, and found that the Δ*crp* strain exhibited markedly reduced hemolytic activity compared with the WT and C-*crp* (Figure 3A) strains. However, there was no difference in the hemolytic activity of the protein extracts from the three strains, and the mRNA levels of hemolytic genes (*vmh* and *tlh*) in wild and deletion strains were equal (Figure 3B,C). We then detected the enzyme activity of the extracellular products of Δ*crp*, WT, and C-*crp*. The lecithinase, protease, urease, and gelatinase activity levels were determined. The results reveal that lower protease, gelatinase, and urease activities (8.333 ± 0.577 mm, 11.333 ± 0.153 mm, and 3.567 ± 0.951 mm) were seen in Δ*crp* compared with those of the wild-type strain (17.000 ± 1.000 mm, 15.400 ± 0.436 mm, and 24.380 ± 1.109 mm) (Figure 3D); the C-*crp* strain restored the enzymatic activity. Nevertheless, the *crp* deletion strain had no significant difference from the wild-type in lecithinase activity. The transcriptional levels of the T2SS genes in the WT, Δ*crp*, and C-*crp* strains were analyzed by qRT-PCR. The qRT-PCR results showed down-regulation of the canonical T2SS operon transcription levels in the deletion strain (Figure 3E), demonstrating that *crp* could play a key role in the T2SS.

### 3.4. Effect of the crp Gene on Colonial Morphology and Biofilm Formation Ability

Lower binding for Δ*crp* was observed using Congo red staining in comparison to WT. A recovered phenotype was seen in the C-*crp* (Figure 4A). Furthermore, WT and C-*crp* showed a smooth colony morphology on the agar plate, while Δ*crp* showed a rugose one (Figure 4A). Additionally, the 96-well plate cultivated with Δ*crp* visibly showed no biofilm when stained with crystal violet, indicating a clear difference from the wild-type strain (Figure 4B). However, the results of the qRT-PCR showed that the transcription levels of the key transcriptional activators of biofilm formation (*vpsR* and *vpsT*) and the biofilm matrix protein genes (*rbmA* and *rbmC*) and IVa pili (*mshA*) were up-regulated in the deletion strain, which indicates that these biofilm factors are controlled negatively by *crp* regulation (Figure 4C). Overall, the loss of the *crp* gene affects colony morphology and impairs bacterial biofilm formation. It is worth noting that the increased production of Vibrio polysaccharides (VPS) is largely responsible for the rugose variant, but our experiment showed the opposite, and the biofilm formation ability also seems to be contrary to the qRT-PCR results.

### 3.5. Effect of the crp Gene on Swimming Motility and Adhesion Ability

Comparing the WT strain to the deletion strain, a different motile phenotype was observed. As shown in Figure 5A, the spreading diameters for WT and Δ*crp* were higher, 2.265 ± 0.062 cm and 0.191 ± 0.030 cm, respectively. We used TEM to assess the flagella synthesis. It was discovered that the absence of flagella causes Δ*crp* to lose motility (Figure 5C).

Adhesion was investigated in vitro in order to assess Δ*crp*’s ability to adhere. The WT strain’s adhesion rate to CCK cells was around 0.398 ± 0.053%, but in the Δ*crp*, it was much lower, at 0.027 ± 0.017% (Figure 5D).

Using qRT-PCR, the transcriptional levels of the flagellin and adhesin genes in the WT, Δ*crp*, and C-*crp* strains were examined. The findings revealed that the majority of the key Class I, II, III, and IV flagellar regulon and adhesin genes were down-regulated, suggesting that *crp* regulation positively controls flagellum and adhesin factors (Figure 5B,E).

### 3.6. Pathogenicity of the crp Deletion Strain in Cells and Animals

Cytotoxicity tests were carried out to evaluate the pathogenic role of ECPs secreted by the WT, Δ*crp*, and C-*crp* strains. The survival rate of Δ*crp* cells at 24 h was found to be substantially higher than that of the WT strain (72.29 ± 11.97% vs. 52.41 ± 6.99%) (Figure 6A). The mRNA level of metalloproteinase gene *hapA* was significantly decreased (Figure 6B. Meanwhile, an injection challenge was conducted. Figure 6C displays the survival curves for each group. The time of death occurred 2–8 days following infection. At 4.00 × 10^6^ and 4.00 × 10^5^, there were significant differences. In addition, the LD_50_ of Δ*crp* was 1.85 × 10^6^ CFU/mL, which is 73.1 times higher than the LD_50_ of the wild type, which is 2.53 × 10^4^ CFU/mL. Therefore, the cumulative mortality via the injection challenge method was considerably reduced by deletion of the *crp* gene.

## 4. Discussion

To investigate the function of *crp* in *V.mimicus*, a *crp* deletion strain was generated using the λ-RED recombination system by natural transformation, following the protocol established by Yu [30]. The strain with the *crp* deletion exhibited a slower growth rate on LB medium, indicating a potential influence of *crp* on metabolic processes. Interestingly, the deletion strain was shown to be incapable of lysine decarboxylation (LDC), ornithine decarboxylation (ODC), and the consumption of carbon sources (dMAL, dMAN, and dTRE), according to its biochemical features (Appendix A). Both LDC and ODC are related to bacterial growth and resistance to external environmental pressure [33,34]. In addition, it has been reported that ODC is related to colonization ability and affects the pathogenicity of *E. coli* (APEC) [35]. However, no evidence is available on Vibrio, which may be a factor in *V. mimicus*’s reduced pathogenicity. The absence of *crp* made the SCCF01 lose the metabolic ability to use maltose, mannitol, and trehalose as carbon sources; the deletion can only use glucose as its sole carbon source. A recent study has also shown that the CRP protein is involved in assessing energy balance within cells and is essential for efficient nutrient assimilation in a competitive environment [36]. Overall, the aspects mentioned above made the Δ*crp* grow slowly. We also found that the *crp* deletion strain had longer cell length. However, this result was contrary to the Δ*crp* strain of *K. pneumoniae* [37]. How *crp* affects cell membranes still needs further study.

The T2SS seen in Gram-negative bacteria consists of vast assemblies that span two membranes and facilitate the secretion of several enzymes or toxins in a folded condition. EpsA to EpsN are the general names for T2SS proteins in *V. cholerae* [38]. The cytosolic EpsE protein is known to be connected with the “secretion ATPase” and forms a complex with a minimum of two bitopic inner membrane T2SS proteins, EpsL, EpsM, and EpsF [39,40]. EpsD, a “secretin” protein, forms a membrane hole for released proteins, and EpsC can precisely regulate this channel [41,42]. This study found that the loss of hemolytic activity of the deletion strain was not due to the decreased synthesis of hemolysin in VMH and TLH, but to the inhibition of hemolysin secretion. Knockout of *crp* significantly inhibited the expression of bacterial T2SS, as well as the secretion of toxins and extracellular products. We can demonstrate that the *crp* gene is a key transcriptional regulator of T2SS in *V. mimicus*. In addition, previous studies in our laboratory have shown that T2SS is a critical virulence factor in *V. mimicus*, and deletion of this gene greatly reduces the virulence of *V. mimicus* [30]. In this experiment, the pathogenicity of Δ*crp* was significantly reduced in both in vitro and in vivo pathogenicity experiments.

The inhibition of T2SS also affected biofilm formation. The main components of *V. cholerae* biofilm are VPS and matrix proteins RbmA and RbmC, and the transcriptional activators *vpsR* and *vpsT* regulate their production [43,44]. IVa pili *mshA* can also promote *V. cholerae* biofilm formation in the initial stage [45]. Although biofilm synthesis genes were upregulated in Δ*crp*, inhibition of T2SS prevented effective secretion of VPS and matrix proteins. The Congo red staining proves this (the ability of Congo red binding shows a positive correlation with the presence of biofilm VPS [46]). The expression levels of *vpsT*, *rbmA*, and *rbmC* in C-*crp* did not recover to SCCF01, possibly because the expression of *crp* in C-*crp* was still different from SCCF01 after induction with arabinose. In conclusion, deletion of the *crp* gene in *V. mimicus* still resulted in reduced biofilm formation and decreased survival ability [47].

The flagella of bacterial pathogens are related to chemotaxis and motility, which is critical for cell invasion. The flagellar regulon of Vibrio is organized into four classes. Class I consists of the *flrA* gene that activates Class II genes’ expression [48]. Class II is composed of the *flrBC*, *fliA*, and *flhA* genes, mostly including genes responsible for structural components, such as the MS ring, and export apparatus components [49]. First found in *V. cholerae*, *flrA*, *flrB*, and *flrC* were shown to be essential for flagellar synthesis [50]. *flrC* and *fliA*, which are found in Class II genes, promote the transcription of class III and class IV genes, respectively [51,52]. Class III genes encode the basal body and hook components; the primary flagellin *flaA* has been shown to be critical for *V. cholerae* motility [53]. Bacteria lacking *crp* are known to exhibit reduced motility [54]. In this study, we found that the swimming motility of Δ*crp* was significantly reduced because the expression of flagella assembly core genes was down-regulated, and the flagella could not be assembled effectively. Thus, we could demonstrate that the *crp* gene regulates flagella formation in *V. mimicus*.

The flagellar assembly pathway also affects bacterial adhesion [55], and our study found that the expression level of the adhesion genes (*acfD* and *ompU*) of Δ*crp* was lower than that of the wild strain. It has been reported that the outer membrane protein (OMP) U and accessory colonization factor (ACF) play important roles in the adhesion process of *V. cholerae* [56,57]. In order to prevent arabinose from impacting cell growth, no arabinose was added to the cell wells to stimulate the exogenous *crp* gene expression of the complementary strain, which might have had an influence on the adhesion experiments. Taken together, the loss of flagella, as well as the downregulation of the adhesion genes, resulted in a decrease in the adhesion ability in Δ*crp*.

Based on all the phenotypes mentioned above, we observed that the *crp* gene affects many factors closely related to bacterial virulence. Moreover, previous studies from our laboratory have shown that the LD_50_ values of *V. mimicus* lacking the T2SS and sialidase are 307-and 27-fold higher, respectively, than that of the wild strain [30,31]. So, considering these, we first conducted a CCK cell viability assay to compare the toxicity of the extracellular products. As a result of the downregulation of the *hapA* gene and the inhibition of the T2SS, the deletion strain had a higher survival rate. The mucinase activity of the Zn-dependent metalloprotease HapA improves the effects of other toxic chemicals and promotes the spread of infected Vibrio; these factors contribute to the pathogenesis of *V. cholerae* [58]. In further pathogenicity assessment assays, the LD_50_ of Δ*crp* in hybrid catfish increased by 73-fold, comparing to the parent strain. This was consistent with the previous study [59]. Together with these findings, the reduced pathogenicity of Δ*crp* can be ascribed to two factors: the first is its weakened capacity for growth; the second is its diminished virulence, which includes inhibition of its adherence, effector delivery system, motility, exoenzyme, and biofilm. These findings show that the *crp* gene is essential to regulating the pathogenicity and virulence of *V. mimicus*.

## 5. Conclusions

In conclusion, our research shows that the *crp* gene is essential for the formation of *V. mimicus* flagella and controls the major virulence factor type II secretion system, adhesion, and metalloproteinase genes. It also plays a role in growth. In the hybrid catfish challenge experiment, the deletion strain clearly showed an attenuation effect. These findings show that the *crp* gene is essential in regulating the pathogenicity and virulence of *V. mimicus*.

## Figures and Tables

**Figure 1 animals-14-00437-f001:**
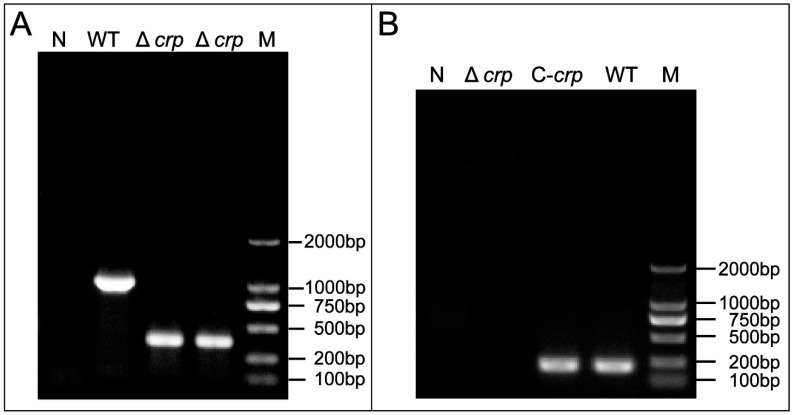
(**A**) Identification of Δ*crp* by PCR using the external region of *crp* gene primers; (**B**) Identification of Δ*crp* and C-*crp* by RT-PCR using the *crp* gene primers; M refers to Marker; N refers to the ddH2O negative control.

**Figure 2 animals-14-00437-f002:**
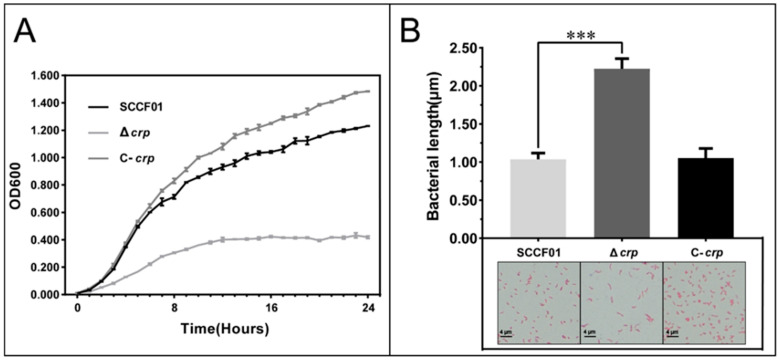
(**A**) Determination of growth curve in LB broth; (**B**) Determination of bacterial length during the logarithmic phase in LB broth (*** *p* < 0.001).

**Figure 3 animals-14-00437-f003:**
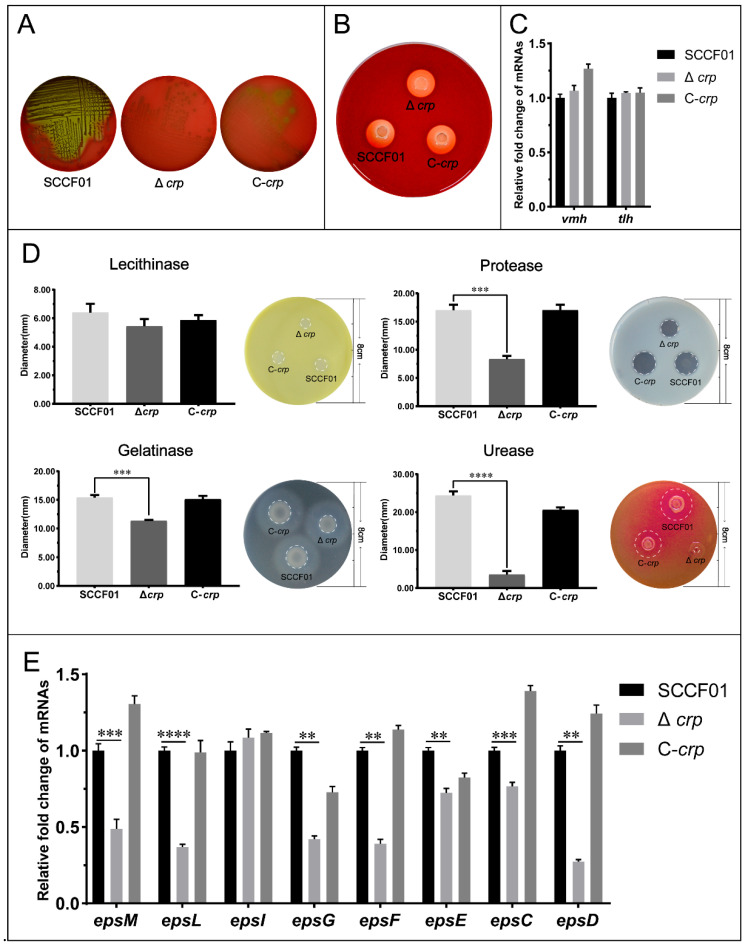
(**A**) Bacterial hemolytic activity assay; (**B**) Bacterial hemolytic activity assay of active proteins extracted from bacteria; (**C**) Quantitative real-time PCR used to determine hemolytic gene transcriptional levels; (**D**) Enzymatic activity assay; (**E**) Quantitative real-time PCR used to determine type II secretion system gene transcriptional levels (** *p* < 0.01, *** *p* < 0.001, and **** *p* < 0.0001).

**Figure 4 animals-14-00437-f004:**
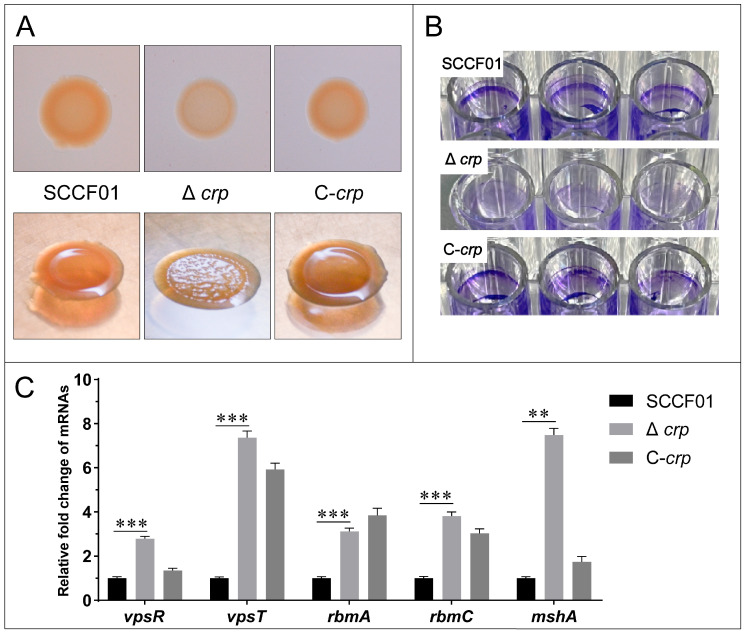
(**A**) Congo red binding and colonial morphology; (**B**) Biofilm formation ability assay; (**C**) Quantitative real-time PCR used to determine the transcriptional levels of the *mshA*, *rbmA*, *rbmC*, *vpsR*, and *vpsT* genes (** *p* < 0.01 and *** *p* < 0.001).

**Figure 5 animals-14-00437-f005:**
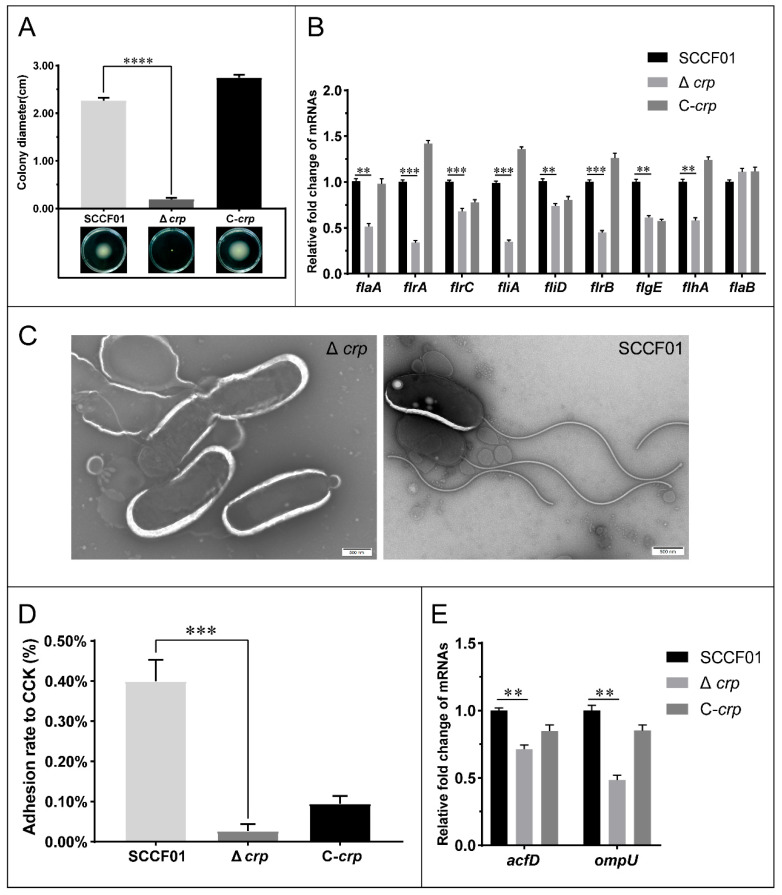
(**A**) Representative swimming motility assays in 0.3% LB agar and measurement of swimming zones; (**B**) Quantitative real-time PCR used to determine the transcriptional levels of flagellin genes; (**C**) Negative staining-transmission electron microscopy of *V.mimicus* SCCF01 and Δ*crp* (scale bar = 500 nm); (**D**) Adhesion ability assay; (**E**) Quantitative real-time PCR used to determine the transcriptional levels of adhesin genes (** *p* < 0.01, *** *p* < 0.001, and **** *p* < 0.0001).

**Figure 6 animals-14-00437-f006:**
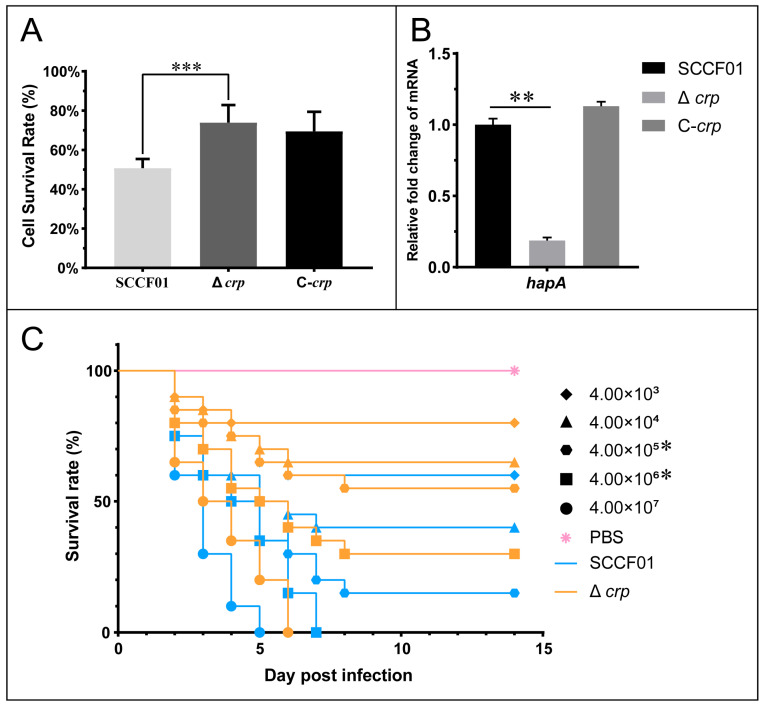
(**A**) Cell survival rate assay; (**B**) Quantitative real-time PCR used to determine the transcriptional levels of the *hapA* gene; (**C**) WT and Δ*crp* survival rate analysis of hybrid catfish (* *p* < 0.05, ** *p* < 0.01, and *** *p* < 0.001).

## Data Availability

Data are contained within the article and Appendix A.

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
