# Peer review of "The cAMP Receptor Protein (CRP) of Vibrio mimicus Regulates Its Bacterial Growth, Type II Secretion System, Flagellum Formation, Adhesion Genes, and Virulence"

_animals, 2024, doi:10.3390/ani14030437_

Round 1

Reviewer 1 Report

Comments and Suggestions for Authors

I appreciate the effort put into this manuscript entitled "cAMP receptor protein (CRP) of Vibrio mimicus regulates bacterial type II secretion system, flagellum formation, adhesion genes and virulence" by Tian et al. Undoubtedly, the study of how crp knockout bacteria effect Vibrio mimicus, a major cause of infectious diseases in Chinese catfish farming, contributes valuable insights to the field. However, I would like to gently suggest that some revisions are necessary before considering it for publication.

Comments:

1. Abstract: Could you elaborate on whether your results can elucidate the primary phenotype impacted by crp? Please provide clarification.

2. Line 36: It would be nice to see the Latin names of prawns and oyster.

3. Line 121: Delete "Colony morphology and ".

4. Line 139: Change "transmission electron microscopy (TEM)" to "transmission electron microscopy (TEM) observation".

5. Line 181: Kindly clarify the parentage of the hybrid catfish. Additionally, it's worth mentioning why hybrid catfish were employed here, considering that SCCF01 was originally isolated from yellow catfish, as mentioned in the introduction.

6. Fig. 1: I noticed there is a lack of relevant data for the complement strain in panel A. Could you please address this issue?

7. Fig. 1A: Only three of the four lanes are marked.

8. Fig. 2: Given the results in panel A, it seems that the crp gene significantly influences the growth of Vibrio mimicus. Could this have an impact on the measurement of other phenotypes in subsequent experiments? For instance, the hemolysis experiment took 96 hours, colony counts took 4 days, biofilm formation took 8 hours, and motility testing took 24 hours.

9. Figure legends: It would be helpful to provide titles for all figures rather than directly stating what A and B represent.

10. Can the deletion strain be stably inherited?

11. I recommend adding a number of new resources on crp gene research to this article's introduction and discussion sections.

12. The article contains occasional non-native English expressions. Please consider revising the entire text to enhance its linguistic quality.

Comments on the Quality of English Language

Minor editing of English language required

Author Response

The point-by-point response to the reviewers' comments (animals-2797153)

Dear editors,

Thank you for your letter and for concerning our manuscript entitled “cAMP receptor protein (CRP) of Vibrio mimicus regulates bacterial type II secretion system, flagellum formation, adhesion genes and virulence” (ID: animals-2797153). The reviewer comments are all valuable and very helpful for revising and improving our paper and the important guiding significance to our manuscript. On behalf of all co-authors, we would like to express our great appreciation to the editor and reviewers and hope that the correction will meet with approval.

The main corrections in the revised manuscript and the response to the reviewer’s comments are as follows:

REVIEWER COMMENTS:

Reviewer #1:

Response to the reviewer: Thanks a lot for the reviewer’s comments. These comments are all valuable and very helpful for revising and improving my paper, as well as the important guiding significance to my research. We have carefully read the comments in the attachment and modified the manuscript one by one. The modifications were shown in the revised manuscript, in red color. Hope the revisions could get your approval.

Question 1#: Abstract: Could you elaborate on whether your results can elucidate the primary phenotype impacted by crp? Please provide clarification.

Response 1#: Thank you for your valuable comments. In this study, the deletion of crp gene resulted in the inhibition of the type II secretion system, resulting in the loss of hemolysis ability, biofilm formation ability and the ability to secrete sufficient extracellular products. The deletion of crp gene resulted in the inhibition of flagella gene and adhesion gene, resulting in the decrease of the swimming ability and adhesion ability. The decreased expression of virulence genes in the deletion strain makes it less virulent. We hope the results can get your approval.

Question 2#: Line 36: It would be nice to see the Latin names of prawns and oyster.

Response 2#: Thank you for your valuable comments. The Latin name for prawn has been added; the Latin name for oyster is not found in the literature. It is possible that V. mimicus is pathogenic to most oysters. We hope the results can get your approval. Change can be found – line 32,33

Question 3#: Line 121: Delete "Colony morphology and ".

Response 3#: Thank you for your valuable comments. Have been corrected. We hope the results can get your approval. Change can be found – line 120

Question 4#: Line 139: Change "transmission electron microscopy (TEM)" to "transmission electron microscopy (TEM) observation".

Response 4#: Thank you for your valuable comments. Have been corrected. We hope the results can get your approval. Change can be found – line 138

Question 5#: Line 181: Kindly clarify the parentage of the hybrid catfish. Additionally, it's worth mentioning why hybrid catfish were employed here, considering that SCCF01 was originally isolated from yellow catfish, as mentioned in the introduction.

Response 5#: Thank you for your valuable comments. The hybrid catfish used was the same as previous animal experiments in our laboratory [1]. Both hybrid catfish and yellow catfish belong to the Siluriformes, and V. mimicus has the same pathogenicity to hybrid catfish [2]. Moreover, the hybrid catfish is a hybrid species bred locally in Sichuan Province, and its disease resistance is stronger than that of yellow catfish, which better indicates the strong virulence of wild strain of V. mimicus. The parentage of the hybrid catfish has been added to the article. We hope the results can get your approval. Change can be found – line 184

Question 6#: Fig. 1: I noticed there is a lack of relevant data for the complement strain in panel A. Could you please address this issue?

Response 6#: Thank you for your valuable comments. In this experiment, the crp gene was constructed on the expression plasmid with T4 ligase, and the plasmid was electrotransformed into the deletion strain. However, the primer used in Figure 1A was designed on the crp gene site of the wild strain, the PCR results of the complementary strain would still be the same as that of the deletion strain, and it could not detect whether the complement was successful or not. In Figure 1B, the expression of crp gene in the wild strain, deletion strain and complementary strain were detected at the RNA level, and it was proved that both deletion and complement experiments were successful. We hope the results can get your approval.

Question 7#: Fig. 1A: Only three of the four lanes are marked.

Response 7#: Thank you for your valuable comments. Have been corrected. We hope the results can get your approval. Change can be found – Fig. 1

Question 8#: Fig. 2: Given the results in panel A, it seems that the crp gene significantly influences the growth of Vibrio mimicus. Could this have an impact on the measurement of other phenotypes in subsequent experiments? For instance, the hemolysis experiment took 96 hours, colony counts took 4 days, biofilm formation took 8 hours, and motility testing took 24 hours.

Response 8#: Thank you for your valuable comments. Before hemolysis, motility, and colony characteristic experiments, the OD600 of the three bacteria in the growth plateau phase were adjusted to the same to avoid the influence of growth [3,4]. In the biofilm experiment, the deletion strain was incubated 8 hours earlier than the wild strain and the complement strain to ensure that the OD600 of the three bacteria were consistent before staining. We hope the results can get your approval.

Question 9#: Figure legends: It would be helpful to provide titles for all figures rather than directly stating what A and B represent.

Response 9#: Thank you for your valuable comments. The plate in this article is to make it easier for readers to read. There is little connection between the small pictures in a plate, so it is hard to list the titles for individual plates. It might be clearer to explain each small figure separately. We hope the results can get your approval.

Question 10#: Can the deletion strain be stably inherited?

Response 10#: Thank you for your valuable comments. We confirmed that the deletion could be stably inherited by PCR identification primer testing after 50 consecutive passages on LB plates. We hope the results can get your approval.

Question 11#: I recommend adding a number of new resources on crp gene research to this article's introduction and discussion sections.

Response 11#: Thank you for your valuable comments. Have been added in the introduction and discussion. We hope the results can get your approval. Change can be found – line 59, 331 to 333

Question 12#: The article contains occasional non-native English expressions. Please consider revising the entire text to enhance its linguistic quality.

Response 12#: Thank you for your valuable comments. We have revised the language throughout the entire paper. We hope the results can get your approval.

Thank you very much for the helpful comments, which greatly improved our manuscript. If any additional edits need to be made or comments addressed, please let us know.

Yours sincerely,

Authors

  1. Zhao, R.; Qin, Z.; Feng, Y.; Geng, Y.; Huang, X.; Ouyang, P.; Chen, D.; Guo, H.; Deng, H.; Fang, J.; et al. Sialic acid catabolism contributes to Vibrio mimicus virulence. Aquaculture 2023, 574, 739660, doi:https://doi.org/10.1016/j.aquaculture.2023.739660.
  2. Geng, Y.; Liu, D.; Han, S.; Zhou, Y.; Wang, K.Y.; Huang, X.L.; Chen, D.F.; Peng, X.; Lai, W.M. Outbreaks of vibriosis associated with Vibrio mimicus in freshwater catfish in China. Aquaculture 2014, 433, 82-84, doi:https://doi.org/10.1016/j.aquaculture.2014.05.053.
  3. Zhou, P.; Han, X.; Ye, X.; Zheng, F.; Yan, T.; Xie, Q.; Zhang, Y.A.; Curtiss Iii, R.; Zhou, Y. Phenotype, Virulence and Immunogenicity of Edwardsiella piscicida Cyclic AMP Receptor Protein (Crp) Mutants in Catfish Host. Microorganisms 2020, 8, doi:10.3390/microorganisms8040517.
  4. Wang, L.; Ling, Y.; Jiang, H.; Qiu, Y.; Qiu, J.; Chen, H.; Yang, R.; Zhou, D. AphA is required for biofilm formation, motility, and virulence in pandemic Vibrio parahaemolyticus. Int J Food Microbiol 2013, 160, 245-251, doi:10.1016/j.ijfoodmicro.2012.11.004.

Reviewer 2 Report

Comments and Suggestions for Authors

The authors constructed a crp gene disruption strain of the highly pathogenic Vibrio mimicus isolated from catfish and measured various virulence genes expression and its virulence. From there, they confirmed that the crp deletion caused a reduction in the expression of many virulence factors, as well as a reduction in virulence itself. A detailed analysis of the role of the crp gene in V. mimicus revealed its central involvement in pathogenicity. These were very important findings for understanding the pathogenicity of V. mimicus.

In p. 2, section 2.1, for plasmids such as pKD4 and CCK cells, please provide the original literature and state the origin.

In p. 3, sections 2.3 and 2.4, the crp mutant strain was very poorly grown, as the authors also mentioned and as can be seen from Figure 2A. Even if OD600 = 1, the phases of growth are different between the wild and mutant strains. It seems that the various activity within the bacteria is quite different between the log phase and the stationary phase. How do you consider that? This is a point that should be considered in other experimental sections as well.

In p. 4, section 2.8, in adhesion to CCK cells, how did you estimate the effect of dead cells due to cytotoxicity as in section 2.10? Were there any cells detached and lost due to cytotoxicity?

In p. 4, section 2.10, please provide a brief explanation of the CCK-8 kit. What kind of substance was being measured, what kind of optical absorption by what wavelengths, etc.?

In p. 7, Fig. 3E, it would be easier to see if the change in expression of the type II secretion apparatus genes due to the crp gene deletion is affected by the overall growth activity of the bacteria by using the house-keeping gene as a control.

In p. 8, Fig. 4C, please discuss why the expression of the vspT, rbmA, and rbmC genes did not restore to the wild-type strain in the complementary strain C-crp.

In p. 9, Fig. 5D, please discuss why adhesion to CCK cells was not restored to the wild-type strain in the complementary strain C-crp.

In p. 10, L312-323, the authors also note that crp gene deletion affect the expression of many basic metabolic genes. Does the decreased expression of house-keeping genes reduce the viability activity of the bacteria, i.e., does it make them more likely to become dead? Was the decreased expression of virulence genes a secondary effect of decreased house-keeping genes expression? A discussion of the correlation with genes that directly act on the expression of virulence factors is needed.

In p. 11, L334, crp gene was thought to be an key regulator of T2SS, but have you confirmed if it was a direct effect?

In p. 11, L374, was the pathogen recovered from catfish inoculated with the crp deletion strain? Was the bacteria died in vivo early after inoculation and were cleared? The susceptibility of the pathogen itself to kill may not be a decrease in virulence.

p. 1, L33, Vibrio cholera -> Vibrio cholerae

p. 6, L219, Maker -> Marker

Gene names throughout of this manuscript should be written with the first letter in lower case. Examples, Fig. 3E, EpsM -> epsM, etc, Fig. 4C, VpsR -> vpsR, etc, p. 11, L350, FlrA -> flrA, etc, etc.

Comments on the Quality of English Language

There were some problems with the notation, but it was generally easy to read.

Author Response

The point-by-point response to the reviewers' comments (animals-2797153)

Dear editors,

Thank you for your letter and for concerning our manuscript entitled “cAMP receptor protein (CRP) of Vibrio mimicus regulates bacterial type II secretion system, flagellum formation, adhesion genes and virulence” (ID: animals-2797153). The reviewer comments are all valuable and very helpful for revising and improving our paper and the important guiding significance to our manuscript. On behalf of all co-authors, we would like to express our great appreciation to the editor and reviewers and hope that the correction will meet with approval.

The main corrections in the revised manuscript and the response to the reviewer’s comments are as follows:

REVIEWER COMMENTS:

Reviewer #1:

Response to the reviewer: Thanks a lot for the reviewer’s comments. These comments are all valuable and very helpful for revising and improving my paper, as well as the important guiding significance to my research. We have carefully read the comments in the attachment and modified the manuscript one by one. The modifications were shown in the revised manuscript, in red color. Hope the revisions could get your approval.

Question 1#: In p. 2, section 2.1, for plasmids such as pKD4 and CCK cells, please provide the original literature and state the origin.

Response 1#: Thank you for your suggestion. We hope the results can get your approval. Change can be found – line 68 to 78

Cell or plasmid

Original literature

Source

Cell

The kidney of channel catfish cells (CCK)

LingBing Zeng, et al. [1]

Jin Xu, et al. [2]

Courtesy of Yangtze River Fisheries Research Institute, Chinese Academy of Fishery Sciences

Plasmid

pKD4

Miaolingbio Co., Ltd

pCP20

Miaolingbio Co., Ltd

pBAD24

Biofeng Biotech Co., Ltd

pKD46

Miaolingbio Co., Ltd

Question 2#: In p. 3, sections 2.3 and 2.4, the crp mutant strain was very poorly grown, as the authors also mentioned and as can be seen from Figure 2A. Even if OD600 = 1, the phases of growth are different between the wild and mutant strains. It seems that the various activity within the bacteria is quite different between the log phase and the stationary phase. How do you consider that? This is a point that should be considered in other experimental sections as well.

Response 2#: Thank you for your valuable comments. During the experiment, we selected single colony of bacteria into test tubes equipped with 10ml fresh LB medium and cultured overnight at 28℃ for 16h. At this time, SCCF01, Δcrp and C-crp all entered the stationary phase. Then we adjusted adjust the bacterial count to OD600 = 1 by dilution and concentration. To sum up, we did all the experiments with bacteria that were in the stationary phase. We hope the results can get your approval.

Question 3#: In p. 4, section 2.8, in adhesion to CCK cells, how did you estimate the effect of dead cells due to cytotoxicity as in section 2.10? Were there any cells detached and lost due to cytotoxicity?

Response 3#: Thank you for your valuable comments. First of all, we refer to the previous method of our laboratory when conducting cell adhesion experiments [3]. Bacteria will be washed three times with sterile PBS solution before adhering to cells to avoid the influence of extracellular products on the experiment. Secondly, bacterial adhesion is the first step in the interaction between bacteria and host, the initial step for bacteria to establish infection, and the premise for bacteria to establish and regulate the structure of bacterial flora in the host. Once bacterial colonization is achieved, its persistence and multiplication can cause damage to the host [4]. During the 30min of the experiment, the bacteria mainly colonized the cells in the first step of the pathogenic process and did not cause serious damage to the cells in a short time. We hope the results can get your approval. Change can be found – line 152 to 154

Question 4#: In p. 4, section 2.10, please provide a brief explanation of the CCK-8 kit. What kind of substance was being measured, what kind of optical absorption by what wavelengths, etc.?

Response 4#: Thank you for your valuable comments. After the cells were incubated with the extracellular products for 24h, 10μl CCK-8 solution was added to each well and left for incubation at 37℃, after incubation for 30 min, the OD450 value was measured and the survival rate of CCK cells was calculated according to the formula. We hope the results can get your approval. Change can be found – line 173 to 175

Question 5#: In p. 7, Fig. 3E, it would be easier to see if the change in expression of the type II secretion apparatus genes due to the crp gene deletion is affected by the overall growth activity of the bacteria by using the house-keeping gene as a control.

Response 5#: Thank you for your valuable comments. Relative genes expression was quantified using the comparative threshold cycle 2−ΔΔCT method with 16S rRNA as the endogenous reference [5]. 16S rRNA is also stably expressed in bacteria. We hope the results can get your approval.

Question 6#: In p. 8, Fig. 4C, please discuss why the expression of the vspT, rbmA, and rbmC genes did not restore to the wild-type strain in the complementary strain C-crp.

Response 6#: Thank you for your valuable comments. In addition to the vpsT, rbmA, and rbmC genes, only a few genes showed the same expression level as the wild strain. We constructed a complementary strain by electrotransformation of a plasmid with crp gene in the deletion strain, and arabinose was added to activate the expression of the plasmid. The expression level of crp gene in the complementary strain was different from that in the wild strain, so this may be the cause of the difference in gene expression. In this article, the flhDC gene expression of the complementary strain did not recover[6]. We hope the results can get your approval. Change can be found – line 359 to 361

Question 7#: In p. 9, Fig. 5D, please discuss why adhesion to CCK cells was not restored to the wild-type strain in the complementary strain C-crp.

Response 7#: Thank you for your valuable comments. Arabinose was required to induce activation of crp gene in complementary strain, the complementary strain was washed three times with sterile PBS solution in the adhesion assay, and arabinose was not added to the cell wells, which may be the reason for the phenotypic difference. The adhesion genes did not fully recover to the level of the wild type strain. We hope the results can get your approval. Change can be found – line 381 to 382

Question 8#: In p. 10, L312-323, the authors also note that crp gene deletion affect the expression of many basic metabolic genes. Does the decreased expression of house-keeping genes reduce the viability activity of the bacteria, i.e., does it make them more likely to become dead? Was the decreased expression of virulence genes a secondary effect of decreased house-keeping genes expression? A discussion of the correlation with genes that directly act on the expression of virulence factors is needed.

Response 8#: Thank you for your valuable comments. cadA and ldcC genes are related to lysine decarboxylation (LDC) [7], and studies have shown that lysine decarboxylase plays diverse roles in the defense response and controls the growth of bacteria [8]. speC and speF genes are related to ornithine decarboxylation (ODC). It has been reported that ODC can promote bacterial growth and help them adapt to or resist external environmental pressure [9], and can enhance the pathogenicity of bacteria in E. coli [10], but there has been no report on the pathogenicity of Vibrio. It is possible that the deletion of crp in V. mimicus not be able to perform ornithine decarboxylation reaction, which may affect the pathogenicity of the bacteria. Both LDC and ODC are related to bacterial survival and growth and resistance to environmental pressure, so the growth rate and resistance to pressure of bacteria after the deletion of crp will be affected, and may be more easily cleared by the external environment than wild strains. We hope the results can get your approval. Change can be found – line 323 to 329

Question 9#: In p. 11, L334, crp gene was thought to be an key regulator of T2SS, but have you confirmed if it was a direct effect?

Response 9#: Thank you for your valuable comments. Our results show that the type II secretion system is suppressed after crp gene deletion, but we do not know whether the regulation is direct or indirect. We hope the results can get your approval.

Question 10#: In p. 11, L374, was the pathogen recovered from catfish inoculated with the crp deletion strain? Was the bacteria died in vivo early after inoculation and were cleared? The susceptibility of the pathogen itself to kill may not be a decrease in virulence.

Response 10#: Thank you for your valuable comments. Study has shown that Δcrp of Edwardsiella piscicida is easier to be cleared by the host after infecting [11], so the decreased pathogenicity of Δcrp to the hybrid catfish in this study may be due to the inhibition of bacterial growth, T2SS, biofilm formation, flagella formation, adhesin synthesis, and virulence gene expression and the decreased ability of anti-host immune clearance. Has been added to the discussion. We hope the results can get your approval. Change can be found – line 395 to 404

Question 11#: p. 1, L33, Vibrio cholera -> Vibrio cholerae

Response 11#: Thank you for your valuable comments. Have been corrected. We hope the results can get your approval. Change can be found – line 31

Question 12#: p. 6, L219, Maker -> Marker

Response 12#: Thank you for your valuable comments. Have been corrected. We hope the results can get your approval. Change can be found – line 227

Question 13#: Gene names throughout of this manuscript should be written with the first letter in lower case. Examples, Fig. 3E, EpsM -> epsM, etc, Fig. 4C, VpsR -> vpsR, etc, p. 11, L350, FlrA -> flrA, etc, etc.

Response 13#: Thank you for your valuable comments. Errors in the article and in the figure have been corrected. We hope the results can get your approval. Change can be found – in the full article

Thank you very much for the helpful comments, which greatly improved our manuscript. If any additional edits need to be made or comments addressed, please let us know.

Yours sincerely,

Authors

  1. ZENG, L.; LI, X.; ZHANG, L.; XU, J.; ZHANG, Y. Establishment and characterization of a cell line derived from kidney of channel catfish, Ictalurus punctatus. Journal of Fishery Sciences of China 2009, 16, 75-81.
  2. Xu, J.; Zeng, L.; Luo, X.; Wang, Y.; Fan, Y.; Gong, S. Reovirus infection emerged in cultured channel catfish, Ictalurus punctatus, in China. Aquaculture 2013, 372-375, 39-44, doi:https://doi.org/10.1016/j.aquaculture.2012.10.011.
  3. Zhao, R.; Qin, Z.; Feng, Y.; Geng, Y.; Huang, X.; Ouyang, P.; Chen, D.; Guo, H.; Deng, H.; Fang, J.; et al. Sialic acid catabolism contributes to Vibrio mimicus virulence. Aquaculture 2023, 574, 739660, doi:https://doi.org/10.1016/j.aquaculture.2023.739660.
  4. Snoussi, M.; Noumi, E.; Hajlaoui, H.; Usai, D.; Sechi, L.A.; Zanetti, S.; Bakhrouf, A. High potential of adhesion to abiotic and biotic materials in fish aquaculture facility by Vibrio alginolyticus strains. J Appl Microbiol 2009, 106, 1591-1599, doi:10.1111/j.1365-2672.2008.04126.x.
  5. Liu, L.; Li, F.; Xu, L.; Wang, J.; Li, M.; Yuan, J.; Wang, H.; Yang, R.; Li, B. Cyclic AMP-CRP Modulates the Cell Morphology of Klebsiella pneumoniae in High-Glucose Environment. Front Microbiol 2019, 10, 2984, doi:10.3389/fmicb.2019.02984.
  6. Tsai, Y.L.; Chien, H.F.; Huang, K.T.; Lin, W.Y.; Liaw, S.J. cAMP receptor protein regulates mouse colonization, motility, fimbria-mediated adhesion, and stress tolerance in uropathogenic Proteus mirabilis. Sci Rep 2017, 7, 7282, doi:10.1038/s41598-017-07304-7.
  7. Krithika, G.; Arunachalam, J.; Priyanka, H.P.; Indulekha, K. The Two Forms of Lysine Decarboxylase; Kinetics and Effect of Expression in Relation to Acid Tolerance Response in E. coli. Journal of Experimental Sciences 2010, 1, 10-21.
  8. Han, L.; Yuan, J.; Ao, X.; Lin, S.; Han, X.; Ye, H. Biochemical Characterization and Phylogenetic Analysis of the Virulence Factor Lysine Decarboxylase From Vibrio vulnificus. Front Microbiol 2018, 9, 3082, doi:10.3389/fmicb.2018.03082.
  9. Park, J.Y.; Kang, B.R.; Ryu, C.M.; Anderson, A.J.; Kim, Y.C. Polyamine is a critical determinant of Pseudomonas chlororaphis O6 for GacS-dependent bacterial cell growth and biocontrol capacity. Mol Plant Pathol 2018, 19, 1257-1266, doi:10.1111/mpp.12610.
  10. Guerra, P.R.; Herrero-Fresno, A.; Pors, S.E.; Ahmed, S.; Wang, D.; Thøfner, I.; Antenucci, F.; Olsen, J.E. The membrane transporter PotE is required for virulence in avian pathogenic Escherichia coli (APEC). Veterinary Microbiology 2018, 216, 38-44, doi:https://doi.org/10.1016/j.vetmic.2018.01.011.
  11. Zhou, P.; Han, X.; Ye, X.; Zheng, F.; Yan, T.; Xie, Q.; Zhang, Y.A.; Curtiss Iii, R.; Zhou, Y. Phenotype, Virulence and Immunogenicity of Edwardsiella piscicida Cyclic AMP Receptor Protein (Crp) Mutants in Catfish Host. Microorganisms 2020, 8, doi:10.3390/microorganisms8040517.

Round 2

Reviewer 2 Report

Comments and Suggestions for Authors

The authors have sincerely answered all of the questions, and the text has been appropriately revised. In addition, please answer to the following questions:

In response to the answer of #1.

The original paper is as follows. Please cite.

KA Datsenko, BL Wanner, One-step inactivation of chromosomal genes in Escherichia coli K-12 using PCR products. Proc Natl Acad Sci USA. 2000 Jun 6;97(12):6640-5. doi: 10.1073/pnas.120163297.

PP Cherepanov, W Wackernagel, Gene disruption in Escherichia coli: TcR and KmR cassettes with the option of Flp-catalyzed excision of the antibiotic-resistance determinant. Gene, 158 (1995), pp. 9-14

In response to the answer of #2.

Bacteria in stationary phase will generally have lower bioactivity. It would not be appropriate to measure pathogenic activity. It is recommended that using bacteria in the logarithmic growth phase and adjusting the turbidity of the bacteria for virulence evaluation.

In response to the answers of #3 and #7.

Was arabinose added to the C-crp strain during culture? When the C-crp strain was cultured with arabinose, all genes should have been expressed to about the same extent as in the wild-type strain, including adhesion factors, as the authors have shown in Figure 5E. Why was there a difference in washing both of them three times with PBS? Discuss a possible setting.

In response to the answer of #4.

There is no description of what the OD450 value is based on. Please indicate the substance name and describe it.

In response to the answer of #5.

Please describe your answer in the manuscript.

In response to the answer of #6.

Even if there was a difference in expression between wild-type and C-crp strains, consider why some genes whose expression was regulated by CRP was equivalent to that of the wild-type strain and some genes whose expression was not.

In response to the answers of #8 and #10.

If crp mutants were susceptible to nutrient depletion conditions, could V. mimicus be considered difficult or impossible to grow in vivo rather than less virulent due to crp mutation? Compared to Fig. 5D and Fig. 6C, Fig. 6B showed no significant differences between the wild-type and crp mutant strains. In particular, there was little difference at 4.00E+07. So, an evaluation based on bacterial challenge on catfish could not make a strong argument for a decrease in virulence. This suggests that the effect of the crp mutation on the virulence of V. mimicus may be limited.

Comments on the Quality of English Language

Ask a native English speaker to correct your English.

Author Response

The point-by-point response to the reviewers' comments (animals-2797153)

Dear editors,

Thank you for your letter and for concerning our manuscript entitled “cAMP receptor protein (CRP) of Vibrio mimicus regulates bacterial growth, type II secretion system, flagellum formation, adhesion genes and virulence” (ID: animals-2797153). The reviewer comments are all valuable and very helpful for revising and improving our paper and the important guiding significance to our manuscript. On behalf of all co-authors, we would like to express our great appreciation to the editor and reviewers and hope that the correction will meet with approval.

The main corrections in the revised manuscript and the response to the reviewer’s comments are as follows:

REVIEWER COMMENTS:

Reviewer:

Response to the reviewer: Thanks a lot for the reviewer’s comments. These comments are all valuable and very helpful for revising and improving my paper, as well as the important guiding significance to my research. We have carefully read the comments in the attachment, modified the manuscript one by one, and asked native English speakers to correct my English. The modifications were shown in the revised manuscript, in RED color. Hope the revisions could get your approval.

Question 1#: In response to the answer of #1. The original paper is as follows. Please cite.

KA Datsenko, BL Wanner, One-step inactivation of chromosomal genes in Escherichia coli K-12 using PCR products. Proc Natl Acad Sci USA. 2000 Jun 6;97(12):6640-5. doi: 10.1073/pnas.120163297.

PP Cherepanov, W Wackernagel, Gene disruption in Escherichia coli: TcR and KmR cassettes with the option of Flp-catalyzed excision of the antibiotic-resistance determinant. Gene, 158 (1995), pp. 9-14

Response 1#: Thank you for your suggestion. Has been added to the article. We hope the results can get your approval. Change can be found – line 70 to 73.

Question 2#: In response to the answer of #2.

Bacteria in stationary phase will generally have lower bioactivity. It would not be appropriate to measure pathogenic activity. It is recommended that using bacteria in the logarithmic growth phase and adjusting the turbidity of the bacteria for virulence evaluation.

Response 2#: Thank you for your valuable comments. Because the phenotypic experiment requires a bacterial solution with OD600=1 and the deletion strain grow slowly, we need to add single colonies of three different strains of bacteria to a 15 ml test tube and culture them at 28 °C for 16 hours prior to the experiment to ensure there are enough bacteria to finish it. As the reviewer said, we will modify the bacteria's turbidity in future experiments. We hope the results can get your approval.

Question 3#: In response to the answers of #3 and #7.

Was arabinose added to the C-crp strain during culture? When the C-crp strain was cultured with arabinose, all genes should have been expressed to about the same extent as in the wild-type strain, including adhesion factors, as the authors have shown in Figure 5E. Why was there a difference in washing both of them three times with PBS? Discuss a possible setting.

Response 3#: Thank you for your valuable comments. Arabinose was added in the culture. Before the adhesion experiment, the bacteria were washed three times with PBS to avoid the influence of extracellular products, and arabinose was also washed away. To avoid the impact of arabinose on cell growth, no arabinose was added to the cell wells to stimulate the complementary strains' exogenous crp gene expression, which might lead to differences in adhesion experiments. We hope the results can get your approval. Change can be found – line 393 to 396

Question 4#: In response to the answer of #4.

There is no description of what the OD450 value is based on. Please indicate the substance name and describe it.

Response 4#: Thank you for your valuable comments. Measure the absorbance at 450 nm using a microplate reader according to the Sangon Biotech (Shanghai) Co., Ltd.'s Cell Counting Kit-8 specification (Product number: E606335).

Description: Cell proliferation assays have been widely used to assess cell cycle regulatory factors such as growth factors, cytokines, mitogens, and drugs. Tetrazolium salts (e.g. MTT, XTT, WST-1 and WST-8) are especially useful for assaying the quantification of viable cells, because they are cleaved to form a formazan dye only by metabolic active cells. Cell Counting Kit-8 (CCK-8) allows sensitive assay for the determination of cell viability in cell proliferation and cytotoxicity assays. WST-8 [2-(2-methoxy-4-nitrophenyl)-3-(4-nitrophenyl)-5-(2, 4-disulfophenyl)-2H-tetrazolium, monosodium salt], the highly water-soluble tetrazolium salt, is reduced by dehydrogenases in cells to give a yellow-color product (formazan), which is soluble in the culture medium. The amount of the formazan dye, generated by the activities of dehydrogenases in cells, is directly proportional to the number of living cells. The detection sensitivity of Cell Counting Kit-8 is higher than assays using other tetrazolium salts such as MTT, XTT, MTS or WST-1. This kit is very convenient for detection.

The kit only contains a tube of CCK-8 solution containing WST-8 that has been prepared, and no preparation and other operations are needed. No isotopes were used, and all procedures were performed in the same 96-well plate. There is no need to wash the cells, no need to collect the cells, and no need to take additional steps to dissolve formazan. It can be used for the detection of large quantities of samples.

We hope the results can get your approval. Change can be found – line 179 to 181

Question 5#: In response to the answer of #5.

Please describe your answer in the manuscript.

Response 5#: Thank you for your valuable comments. It has been mentioned in the manuscript and is marked in red. We hope the results can get your approval. Change can be found – line 207 to 208

Question 6#: In response to the answer of #6.

Even if there was a difference in expression between wild-type and C-crp strains, consider why some genes whose expression was regulated by CRP was equivalent to that of the wild-type strain and some genes whose expression was not.

Response 6#: Thank you for your valuable comments. It is true that in our experiment, several genes, including those related to the type II secretion system, biofilm genes, and some flagella, could not be expressed at the same level as in the wild strain. This has been shown in other study as well, but the authors do not explain it [1]. This problem also confuses and concerns us, and we will investigate in it. We hope the results can get your approval.

Question 7#: In response to the answers of #8 and #10.

If crp mutants were susceptible to nutrient depletion conditions, could V. mimicus be considered difficult or impossible to grow in vivo rather than less virulent due to crp mutation? Compared to Fig. 5D and Fig. 6C, Fig. 6B showed no significant differences between the wild-type and crp mutant strains. In particular, there was little difference at 4.00E+07. So, an evaluation based on bacterial challenge on catfish could not make a strong argument for a decrease in virulence. This suggests that the effect of the crp mutation on the virulence of V. mimicus may be limited.

Response 7#: Thank you for your valuable comments. In Fig. 6B, the 73.1-fold attenuation after deletion of crp gene is a significant difference. Moreover, there was a difference at 4.00E+06 and 4.00E+05 (p < 0.05).

The deletion strain's growth rate was lowered, as the reviewer noted, and this slower growth rate also contributed to the strain's decreased pathogenicity toward the host. But it cannot be denied that, this experiment has demonstrated that the deletion strain has reduced virulence, these include adherence, effector delivery system, motility, exoenzyme, and biofilm. The effect of crp gene on the virulence of Vibrio mimicus is significant. The challenge experiments on hybrid catfish were carried out to prove the reduced pathogenicity of strains. In conclusion, both the deletion strain's slower growth and lower virulence contribute to its decreased pathogenicity to hybrid catfish. Change can be found – section 3.6, Fig.6, line 410-424, section 5, title

Thank you very much for the helpful comments, which greatly improved our manuscript. If any additional edits need to be made or comments addressed, please let us know.

Yours sincerely,

Authors

  1. Hassan, H.A.; Ding, X.; Zhang, X.; Zhu, G. Fish borne Edwardsiella tarda eha involved in the bacterial biofilm formation, hemolytic activity, adhesion capability and pathogenicity. Arch Microbiol 2020, 202, 835-842, doi:10.1007/s00203-019-01794-x.
